# Comprehensive Transcriptome Reveals an Opposite Regulatory Effect of Plant Growth Retardants in Controlling Seedling Overgrowth between Roots and Shoots

**DOI:** 10.3390/ijms20133307

**Published:** 2019-07-05

**Authors:** Yanhai Ji, Guanxing Chen, Xuyang Zheng, Qiwen Zhong, Mingyun Zhang, Zhanhui Wu, Changlong Wen, Mingchi Liu

**Affiliations:** 1Beijing Vegetable Research Center, Beijing Academy of Agriculture and Forestry Sciences, Beijing 100097, China; 2Key Lab of Urban Agriculture (Northern), Ministry of Agriculture and Rural Affairs, Beijing 100097, China; 3Academy of Agriculture and Forestry Sciences, Qinghai University, Xining 810016, China

**Keywords:** plant growth retardants, gibberellins, overgrowth, transcriptome, tomato

## Abstract

Seedling overgrowth always develops in undernourished plants due to biotic or abiotic stresses, which significantly decrease the yield of crops and vegetables. It is known that the plant growth retardants paclobutrazol (PBZ) and chlormequat chloride (CCC) are the most commonly used chemicals in controlling seedling height in plants by regulating the gibberellin (GA) biosynthesis pathway. However, the exact molecular regulation mechanism remains largely unknown. This study performed a comprehensive transcriptome profile to identify significantly differentially expressed genes after adding CCC and PBZ to the water culture seedling raising system for the first time. According to the obviously restrained shoots and roots, the GA biosynthesis genes were significantly decreased, as well as the endogenous GA content being reduced. Intriguingly, the GA signaling pathway genes were affected in opposite ways, increasing in roots but decreasing in shoots, especially regarding the phytochrome interacting factor *SlPIF1* and the downstream genes expansins (*SlEXPs*), which promote cell wall remodeling. Further study found that the most down-regulated genes *SlEXPA5* and *SlEXPA15* were expressed specifically in shoot tissue, performing the function of repressing elongation, while the up-regulated genes *SlEXPB2* and *SlEXPB8* were proven to be root-specific expressed genes, which may promote horizontal elongation in roots. This research reported the comprehensive transcriptome profiling of plant growth retardants in controlling seedling overgrowth and restraining GA biosynthesis through the regulation of the GA signaling-related genes *SlPIF1* and *SlEXPs*, with an opposite expression pattern between roots and shoots.

## 1. Introduction

Seedling overgrowth is an important factor influencing the production of many crops and vegetables. In recent years, vegetable seedlings demand has exceeded 300 billion in China, of which traded tomato seedlings made up about 3–4 billon. However, vegetable seedlings of tomato, cucumber, and eggplant often grow excessively under weak light, high temperature, and high humidity conditions, leading to weak growth and decreased yield in the process of cultivation. Previous research reported that the overgrowth of seedlings in the early period of tomato growth could reduce the total output by more than 30% [1]. Thus, suppressing the overgrowth of seedlings is an important issue in the raising of vegetable seedlings, and several methods of adjusting the diurnal temperature [2], shortening the sunshine [3,4], or using brush and impedance have been well utilized [5,6]. Moreover, through chemical methods, plant growth retardants were proven to be the most effective way to inhibit overgrowth. However, the usage of plant growth retardants in the water culture seedling raising system was seldom reported, as well as the molecular regulation of controlling seedling overgrowth remaining unclear.

The plant growth retardants PBZ and CCC are commonly utilized to control plant height and to increase the number of pods produced per plant, as well as to restrict seedling overgrowth [7,8,9,10]. It was reported that PBZ and CCC could affect levels of hormones in regulating floral induction and differentiation [11]. Furthermore, the PBZ was shown to be a synthetic inhibitor of the activity of ent-kaurene oxidase (KO), which was the enzyme catalyzing the oxidation of ent-kaurene to ent-kaurenoic acid in the GA biosynthesis pathway [12,13]. In addition, the CCC could inhibit the activity of ent-kaurene synthase (KS), the key enzyme catalyzing geranylgeranylpyrophosphate (GGPP) to ent-kaurene in GA biosynthesis [13]. These GA biosynthesis genes were associated with plant growth and development [14,15,16,17], due to the central regulation role of GA [18,19,20]. In addition, the GA signaling pathway members were reported to control plant growth and development, including the *DELLA*, phytochrome interacting factors (PIFs), PBZ resistance (PERs), and the downstream cell expansion family genes [21,22,23,24,25]. 

Plant growth and development are regulated at the cellular level by multiple phytohormones, especially GA, which is well studied to control plant elongation [26]. Recently, transcriptome research (also called RNA-Seq) has permitted a robust assessment of the transcript structure and abundance in GA regulating plant growth [27]. Therefore, transcriptome profiling analysis in plant growth retardants affecting GA metabolism remained unknown worldwide, especially in the water culture seedling raising system. In this study, we performed a comprehensive transcriptome analysis to identify genes that were significantly differentially expressed after CCC and PBZ treatment in order to reveal the molecular regulation mechanism of plant growth retardants in controlling seedling overgrowth. Our objective was to draw a comprehensive transcriptome profile of the regulation pathway of CCC and PBZ treatments affecting seedling development, especially in the regulation of GA metabolism and signaling pathways, as well as revealing the cross talk between the plant growth retardants and plant hormone GA in controlling tomato seedling overgrowth. This study also explored the key GA signaling genes in restraining shoot and root growth, such as the *SlPIF1* and the downstream genes *SlEXPA5*, *SlEXPA15*, *SlEXPB2*, and *SlEXPB8*, which restricted the elongation of aboveground shoots and enlarged the underground root diameter.

## 2. Results

### 2.1. CCC and PBZ Treatments Restrain Tomato Seedlings Overgrowth

The morphological features and dynamic changes in developing tomato seedlings are shown in Figure 1. Seedlings under CCC and PBZ conditions displayed significant morphological changes. In general, seedling growth, including hypocotyl, plant height, stem diameter, and root length, was significantly reduced under the CCC and PBZ treatments in the entire seedling growth period (Figure 1). As is shown in Appendix A, the hypocotyl, plant height, stem diameter, and root length were decreased significantly (*p* < 0.01) in the three stages (two-leaf stage, three-leaf stage, and four-leaf stage). For example, the plant height was about 8 cm at the two-leaf stage and increased to 18 cm at the three-leaf stage, while under the CCC and PBZ treatments, the plant height was increased from 2.7 cm and 2.6 cm to 3.6 cm and 3.7 cm, respectively (Appendix A). 

Despite the similar restrained seedling growth of CCC and PBZ, they had a different regulation power in terms of limiting seedling development in some organs, such as in hypocotyl and root length. The CCC and PBZ treatment showed more of an effect on root length; this was much shorter, at 4.69, 10.1, and 8.45 cm, compared with that in CK (17.13, 17.35, and 18.55 cm), and in the PBZ treatment (10.02, 10.1, and 11 cm) from the two-leaf stage to four-leaf stage (Figure 1, Appendix A), while PBZ restrained more hypocotyl growth than the CCC treatment (Appendix A). These results indicated that CCC and PBZ had efficient regulation in water culture system, but had different regulation mechanisms in controlling seedling growth.

### 2.2. CCC and PBZ Increased the Root Diameter in Tomato Seedlings

The tomato seedling roots at the three-leaf stage were observed by a cytohistological experiment, and the root diameter was increased after CCC and PBZ treatment, with PBZ affecting the root diameter much more strongly than CCC (Figure 2). We found that the number of cells around the medulla remained as nine layers, and they were the same layers among CK and CCC, PBZ treatment roots, but the cell size was significantly increased after being fed with plant growth retardants. In addition, the medulla was larger in the CCC/PBZ treatment than in CK, as shown in Figure 2. This indicated that CCC and PBZ could expand the root elongation in an orientation way. 

### 2.3. Transcriptome Profiling Analysis in CCC/PBZ Controlling Tomato Overgrowth

The expression profiling of roots and shoots under CCC and PBZ treatments was performed at the three-leaf stage using mRNA-Seq. For each material, three biological replicates were sequenced. The total number of reads produced for each sample ranged from 36 to 47 million reads. Reads were mapped to the tomato genome, which was generated by the assembly of next-generation reads; in total, the tomato genome encompassed 67.9 Mb, with 34,727 protein-coding genes and 34,727 gene models. Of the total reads generated, for each sample, approximately 30–39 million (76–86% of the total) mapped to the tomato genome. To assess the experimental variation, the gene expression pattern was analyzed and high levels of Pearson’s correlation coefficient (0.88) were observed (Appendix A), which indicated that the experimental methods of sampling and analysis were robust. 

RNA-Seq is a powerful and efficient tool for large-scale gene expression analysis. In this study, the transcriptome analysis indicated that 1132 and 1371 genes were significantly differentially expressed under CCC and PBZ treatments in shoots, respectively. Additionally, 3028 genes were co-expressed differentially after CCC and PBZ treatments (Figure 3, Appendix A). In the roots, 583 and 1924 genes were significantly differentially expressed under CCC and PBZ treatments, and 464 were co-expressed differentially (Figure 3, Appendix A). Moreover, we found that more differentially expressed genes were obtained in shoot than in root; this explains that the shoot growth was identically limited more than in roots (Figure 1). Additionally, more differentially expressed genes were obtained in PBZ-treated roots than in CCC; this demonstrates that PBZ led to more cell expanding than CCC (Figure 2). In addition, we found that the CCC, PBZ, and CCC/PBZ coordinately regulated 127 and 126 key genes in the shoot and root (Figure 3), and those genes played crucial role in restraining the aboveground and underground growth, as well as controlling the overgrowth in tomato seedlings.

### 2.4. Plant Growth Retardants Induced Differentially Expressed Genes in GA Metabolism and the Signaling Pathway 

As CCC and PBZ can inhibit specific steps of GA biosynthesis, this study analyzed the key genes involved in the metabolism and signaling pathways of GA. The genes involved in GA metabolism and signaling pathways showed obvious different regulations both in roots and shoots under the CCC and PBZ treatments, The GA metabolism pathways genes *GA20ox* or *GA3ox* were significantly decreased in CCC/PBZ-treated roots and PBZ-treated shoots, and the degradation gene *GA2ox* was also down-regulated in CCC/PBZ-treated roots and CCC-treated shoots (Figure 4). Intriguingly, the key genes involved in GA signaling pathways were differentially expressed, such as *DELLA*, *PIF1*, PREs, and an amount of expansin genes *SlEXPs* (such as *SlEXPA5*, 7, 8, 14, and 15, and *SlEXPB*2, 8) after feeding with CCC and PBZ were up-regulated in roots but down-regulated in shoots. Additionally, these genes were validated by qRT-PCR technology (Figure 4 and Figure 5). These GA signaling pathway genes displayed an opposite regulation direction between underground roots and aboveground shoots, which may uncover the molecular mechanism of CCC and PBZ in controlling tomato seedling overgrowth.

The negative regulator phytochrome interacting factor *SlPIF1* was increased in CCC/PBZ-treated root and was decreased both in CCC/PBZ-treated shoots (Figure 4, Figure 5A,B). Because of that, the *SlPIFs* could restrain plant growth in the dark and promote plant growth in the light, and the increased *SlPIF1* in underground roots would lead to short roots by restraining root growth and development after plant growth retardant treatment, while the decreased *SlPIF1* in aboveground shoots was against shoot growth and led to dwarf plant height (Figure 1, Figure 4 and Figure 5). Moreover, we observed that an amount of expansin genes *SlEXPs* were unregulated in roots but down-regulated in shoots under CCC and PBZ treatment; this indicates that *SlEXPs* genes may play critical roles in restraining tomato overgrowth both in underground roots and aboveground shoots (Figure 4). Additionally, their expression pattern was validated by qRT-PCR technology (Figure 5). In conclusion, the plant growth retardants induced GA signaling pathway genes that were expressed in an opposite pattern between underground roots and aboveground shoots, which may be according to the restricted shoot/root growth but the expanded root diameter (Figure 1 and Figure 2).

### 2.5. Cross Talk of Plant Growth Retardants and Plant Hormone GA in Controlling Overgrowth

The GA levels in tomato seedling roots and shoots at the three-leaf stage were determined after CCC and PBZ treatment by enzyme-linked immunosorbent analysis (ELISA). The GA content was shown to be significantly reduced in roots, and PBZ caused more GA content to decrease than CCC did, which was consist with the GA metabolism genes *GA20ox* and *GA3ox* being down-regulated in transcriptome profiling (Figure 4 and Figure 6). In the shoots, the GA content was slightly influenced by CCC and PBZ treatment, and the CCC led to more GA dropping then PBZ did; this result was in line with gene expression changes in transcriptome profiling (Figure 4 and Figure 6). The plant hormone GA was much more affected in roots than in shoots after being fed with the plant growth retardants CCC and PBZ, which may be caused by the water culture system in this experiment.

In this study, the CCC and PBZ treatment restrained the endogenous GA content in tomato roots and shoots, as well as the metabolism and signaling pathway genes being differentially expressed, and it was combined with the different shoot and root phenotype alternation led by two independent plant growth retardants, which indicated that a cross talk mechanism existed between plant growth retardants and plant hormones. Consequently, our result demonstrated that the plant growth retardants would be in coordination with plant hormone GA in controlling tomato seedlings’ growth and development, both in the underground roots and aboveground shoots, and the plant growth retardants CCC and PBZ were upstream of the plant hormone GA in controlling seedling overgrowth in tomato, despite the cross-talk mechanism between plant growth retardants and hormones further needing to be uncovered.

### 2.6. Plant Growth Retardants Caused an Opposite Regulation of Tissue-Specific SlEXPs Genes in Restraining Tomato Seedling Overgrowth

The transcriptome profiling observed several expansin genes *SlEXPs* showed different regulation directions between roots and shoots: they were increased in roots but decreased in shoots after being fed with plant growth retardants CCC and PBZ (Figure 4, Appendix A). In order to illustrate the expression pattern of *SlEXPs* genes in tomato, especially the root and shoot tissue-specific expression pattern, this study applied a comprehensive expansin gene discovery method based on a previous instruction [28]. A total of 41 tomato expansin genes were identified in the tomato genome and were specifically grouped into four subfamilies—α-expansin (*SlEXPA*1-27), β-expansin (*SlEXPB*1-8), expansin-like A (*SlEXLA*1-3), and expansin-like B (*SlEXLB*1-4)—as well as being given a name according to that in Arabidopsis and the MEGA analysis result (Appendix A). Finally, we reanalyzed the transcriptome data and obtained 31 *SlEXPs* gene expression datasets in the transcriptome profiling, and also observed the expression pattern in roots and shoots under plant growth retardant CCC and PBZ treatment (Figure 7A). 

The heatmap in Figure 7 illustrates the 31 *SlEXPs* genes’ tissue-specific expression patterns, as well as the regulation under CCC and PBZ treatment in tomato seedlings; we found that 16 *SlEXPs* genes (*SlEXPA*1, 4, 7, 8, 9, 16, 17, 18, 19, 24, and 27; *SlEXPB*1, 2, 4, and 8; and *SlEXLB1*, 4) expressed specificity in roots compared with shoots, and eight *SlEXPs* genes (*SlEXPA*2, 5, 11, 14, 15, and 26; *SlEXPB1;* and *SlEXLB*2) expressed specificity in shoots rather than in roots (Figure 7A). Of these, 11 were root-specific expressed *SlEXPs* genes (*SlEXPA*7, 8, 16, 17, 18, 19, 21, 24, and 27; *SlEXPB*2, 8) and they were up-regulated under CCC and PBZ treatment, while seven shoot-specific *SlEXPs* genes (*SlEXPA*5, 11, 14, 15, and 26; *SlEXPB1*; and *SlEXLB*2) were down-regulated after plant growth retardant treatment (Figure 7A). Moreover, the *EXPA*5, 8, 15, and 26 and *SlyEXPB*2, 8 were confirmed by qRT-PCR experiment (Figure 7B,C). Despite the root-specific *SlyEXPA1* being significantly decreased and the shoot-specific *SlEXPA2* being significantly increased under CCC and PBZ treatment, we concluded that the root-specific expressed *SlEXPs* were up-regulated whereas the shoot-specific expressed *SlEXPs* were down-regulated by plant growth retardants CCC and PBZ in controlling overgrowth in tomato seedlings.

## 3. Discussion

### 3.1. Plant Growth Retardants Restrain GA Biosynthesis in Controlling Overgrowth

It has been well studied that CCC and PBZ are important plant growth retardants which restrict GA biosynthesis and plant growth and development [12,13]. In this study, the seedling growth, including shoot and root length, was significantly reduced under the CCC and PBZ treatments; however, the root diameter was increased. The previous reports demonstrated that CCC and PBZ could affect plant hormone GA biosynthesis by restraining *KS* and *KO* genes in the metabolism pathway [12,13]. The present study found that key genes in the GA biosynthesis *GA20ox*, *GA3ox,* and *GA2ox* were down-regulated in the root rather than the *KS* and *KO* genes, and the PBZ led to more of a decrease in these genes’ expression. This difference may be due to the water culture system used and the three-leaves stage test, of which the test time point was about 20 days post CCC/PBZ treatment. In any case, the GA biosynthesis after treatment with CCC/PBZ was restrained because the endogenous GA content was reduced significantly, and the PBZ treatment reduced the GA content more than CCC. Therefore, the plant growth retardants CCC/PBZ could restrain GA biosynthesis in controlling overgrowth, and this was evidence of shortening shoots and roots, despite the reduced down-regulation of *GA20ox* and *GA2ox* in GA biosynthesis. This was consistent with previous studies that the plant growth retardants were inhibitors of GA, and similar results were obtained in recent research [29,30,31]. 

### 3.2. Plant Growth Retardants Regulate GA Signaling in an Opposite Way between Roots and Shoots

Interestingly, this study observed that GA signaling pathway genes were significantly regulated both in roots and shoots under CCC/PBZ treatment, especially the *SlPIF1* and the downstream genes of *SlEXPs*. This indicated that the plant growth retardants CCC and PBZ could not only restrain the GA signaling pathway, but also could restrict the GA signaling pathway. The expression of *SlPIF1* in roots was increasingly affected after being fed with CCC/PBZ, while the *SlPIF1* in shoots was down-regulated. This result was consistent with its function in growth and germination both in the dark and in the light, because it was sensitive to light [32,33]. Additionally, this finding indicated an opposite regulation mechanism of plant growth retardants in controlling seedling overgrowth. Due to the key roles in promoting plant growth and development of *SlPIF1*, its potential targets genes were investigated in the transcriptome dataset [32,33,34]. Thus, we assumed that the key gene *SlPIF1* was a critical factor in controlling the GA signaling pathway, as well as in restraining seedling overgrowth. In this study, a number of downstream genes of the GA signaling pathway, such as the expansin genes *SlEXPs,* was observed to be co-expressing with *SlPIF1* both in roots and shoots, and these *SlEXPs* genes may be potential transcriptional regulated targets of *SlPIF1*, which was well investigated in previous studies [32,33,34,35,36]. Moreover, we observed that the promoters of these co-expressed *SlEXPs* harbored the G-box element (Appendix A), which was the typical binding site of the transcription factor SlPIF1. Despite this, these transcriptional regulations will be further clarified in the next steps.

### 3.3. Plant Growth Retardants Control Overgrowth through the Tissue-Specific SlEXPs Genes 

It is well known that the *SlEXPs* genes mainly promote cell wall remodeling and increase the volume of a cell [32,33,34,35,36]. This study obtained numbers of *SlEXPs* genes which were influenced by plant growth retardants CCC and PBZ, of which some were up-regulated in underground roots and were down-regulated in the aboveground shoots. Intriguingly, we found that most of the increased *SlEXPs* genes were specifically expressed in root tissues, and most of the decreased *SlEXPs* genes were expressed specifically in the shoots. Moreover, the decreased *SlEXPs* genes in shoots were consistent with the restricted length, but the increased *SlEXPs* genes in roots seemed to disagree with the short roots obtained. However, we found that the root diameter was increased and the cell volume in roots was increasing; this indicates that the up-regulated genes may function in promoting horizontal elongation. Therefore, we built a schematic model of the plant growth retardants CCC and PBZ in controlling tomato seedling overgrowth, which was divided into the aboveground and underground parts; in addition to that, the aboveground and underground parts may influence each other (Appendix A). Therefore, this hypothesis requires more evidence in upcoming studies, especially regarding the validation of transformation in tomato seedlings. 

## 4. Materials and Methods 

### 4.1. Plant Materials

Yingfen No. 8 (*Solanum lycopersicum* L.) tomato seeds were used in this study, which is one of the most important cultivated varieties in China harboring the pink tomato; they were obtained from the Beijing Vegetable Research Center. The pink tomato varieties were extensively planted for more than 50% of cultivation in China. The seeds of Yingfen No.8 were soaked in water for 24 h to promote germination, then transferred to dedicated cultivation trays and put in nutrient solution (5 mM KNO_3_, 5 mM Ca(NO_3_)_2_, 2 mM MgSO_4_, 1 mM KH_2_PO_4,_ 50 M FeNa_2_(EDTA)_2_, 50 M H_3_BO_3_, 10 M MnC_12_, 0.8 M ZnSO_4_, 0.4 M CuSO_4_, and 0.02 M (NH_4_)_6_MoO_24_) after germination. Then, the tomato seedlings were cultivated in the greenhouse at a photoperiod light/dark of 16/8 h, with temperatures during the day/night of 28–32 °C/18–22 °C. The treatments of CCC (20 ppm) and PBZ (1 ppm) were added into the water-cultured nutrient solution, based on the well-chosen concentrations in the pre-experiment. This cultivation condition was proved as be the most effective method of tomato seedling nursing established by Beijing Vegetable Research Center, China. The PBZ/CCC treatments and control groups included five biological replicates.

### 4.2. Cytohistological Observations of Seedling Roots after PBZ and CCC Treatment

Because of the water culture system used in this study, the seedling roots at the three-leaf stage were collected and fixed into the formalin, acetic acid, and alcohol (FAA) for 24 h. Then, they were washed twice with dH_2_O for 15 min, and dehydrated through the graded ethanol solution for 60 min (from 50%, 70%, 85%, 90%, 95%, to 100%). After this, they were gradually infiltrated and embedded with paraffin, sliced into transverse sections by microtome, stained, and observed using a microscope (DMLB, Leica, Wetzlar, Germany). 

### 4.3. Total RNA Extraction and RNA-Seq Library Construction and Sequencing

The three-leaf stage tomato seedlings of Yingfen No.8 were analyzed by transcriptome technology. Total RNA was extracted using the Tiangen RNA extraction kit following the manufacturer’s instructions (Tiangen Biotech, Beijing, China). After measuring the quality in the NanoDrop 1000 spectrophotometer (Thermo Fisher Scientific, Waltham, MA, USA) and on the 1% non-denaturing agarose gel, the strand-specific RNA-Seq libraries were constructed following the previously reported protocol described by Zhong et al. [37]. Then, they were sequenced on the Illumina HiSeq 4000, and three biological replicates for each treatment were performed. The raw data were deposited in NBCI (accession numbers SRP012849 and SRP051354).

### 4.4. Differentially Expressed Genes Affected by PBZ and CCC Treatment

The transcription raw reads were filtered by Trimmomatic software and aligned to the ribosome RNA (rRNA) database using Bowtie, then aligned to the tomato genome sequences using Tophat with the given parameters [34]. To identify differentially expressed genes (DEGs), the normalized reads per kilobase per million mapped reads (RPKM) were calculated, and the Variance Stabilized Data module of DESeq was used. F-tests and *p*-values were analyzed using the Benjamini–Hochberg procedure, subsequently. The GO term enrichment of DEGs was analyzed using the software GO:TermFinder, and the significant DEGs in the GA biosynthesis and metabolism pathways were identified using the Plant MetGenMAP system. 

### 4.5. Validation of Significantly Differentially Expressed Genes

The transcriptional expression patterns of these genes were detected using qRT-PCR with minor modifications in the three-leaf stage shoots and roots, according to the PCR procedure of initial incubation at 94 °C for 3 min, and 40 cycles of denaturation at 94 °C for 15 s, then hybridization at 58 °C for 15 s, as well as extension at 72 °C for 20 s. The complete set of primer sequences is shown in the Appendix A, and the efficiency of the primers was determined by means of standard curves. The efficiency ranged from 90–110% (Appendix A) and the R^2^ values (coefficients of determination) were higher than 0.993 for each gene (Appendix A). Three replicates were performed in this study.

To comprehensively observe the differential expression of important GA pathway downstream expansin genes, the genome-wide expansin genes were blasted and confirmed by using Basic Local Alignment Search Tool (BLAST) and a database Phytozome database (http://www.phytozome.com). Additionally, a phylogenetic tree was performed by ClustalW software and MEGA 4.0 software with NJ method with a Kimura 2-parameter model. qRT-PCR was applied according to the above instructions.

### 4.6. Quantification of Endogenous GA Content by Enzyme-Linked Immunosorbent Assay (ELISA)

The endogenous GA of tomato seedlings in the CK and PBZ/CCC treatments was extracted in ice-cold phosphate-buffered saline solution (PBS, pH 7.4), and the supernatant was extracted using centrifugation at 4000 *g* (4 °C) for 20 min, then it was stored in a refrigerator at −20 °C for the ELISA test. ELISA was performed on a 96-well microtitration plate, as described previously [38]. Three biological replicates were set up.

## 5. Conclusions

The plant growth retardants PBZ and CCC were used to control overgrowth in water-cultured tomato seedlings, with concentrations of 1 ppm and 20 ppm, respectively. The plant growth retardants PBZ and CCC could restrain the expression of GA synthesis and signaling gene expression, but with an opposite regulation between the shoots and roots. With the DEGs related to the GA pathway, the *SlPIF1* and the downstream genes *SlEXPs* were revealed to be key regulators in controlling overgrowth, as well as markers genes for evaluating plant growth in tomato seedlings.

## Figures and Tables

**Figure 1 ijms-20-03307-f001:**
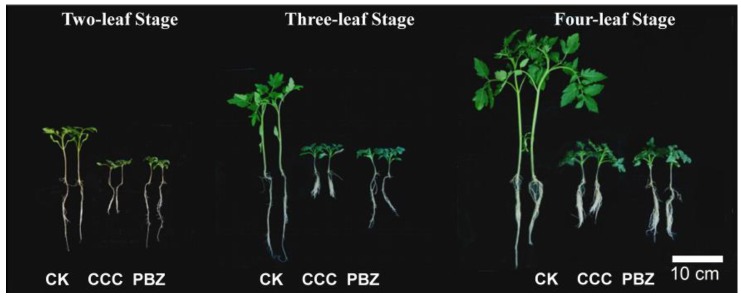
Seedling morphology observations and comparisons under chlormequat chloride (CCC), paclobutrazol (PBZ) treatments, and in the control (CK). The tomato seedling morphology was observed in the two-leaf stage, three-leaf stage, and four-leaf stage; the plant growth retardants CCC and PBZ restrict growth both in the roots and shoots, and the CCC treatment displayed more significant morphological changes.

**Figure 2 ijms-20-03307-f002:**
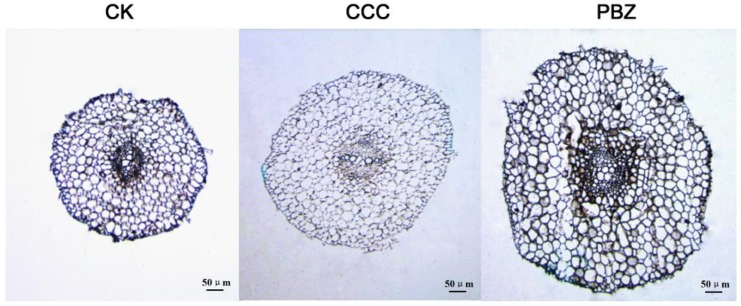
Root diameter observation in tomato roots under CK, CCC and PBZ treatments. The plant growth retardants CCC and PBZ raised the root diameter, and the PBZ treatment induced more orientation elongation in roots than that of CCC treatment, due to the cell size being significantly increased.

**Figure 3 ijms-20-03307-f003:**
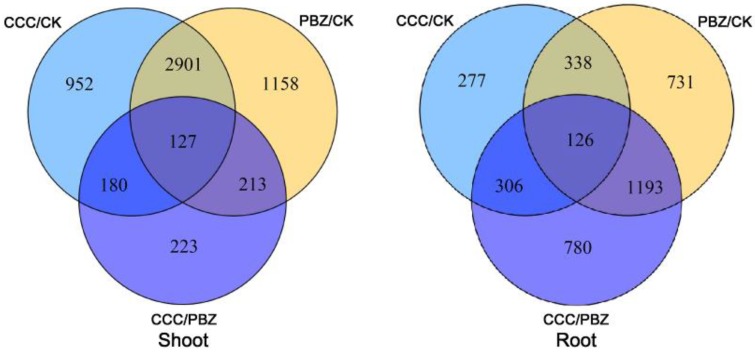
Transcriptome profiling analysis under CK, CCC, and PBZ treatments in root and shoot, respectively. The plant growth retardants CCC and PBZ induced 1132 and 1371 differentially expressed genes in the shoots, and 583 and 1924 differentially expressed genes in the roots of tomato seedlings at the three-leaf stage. The PBZ treatment induced more differentially expressed genes in roots than the CCC treatment.

**Figure 4 ijms-20-03307-f004:**
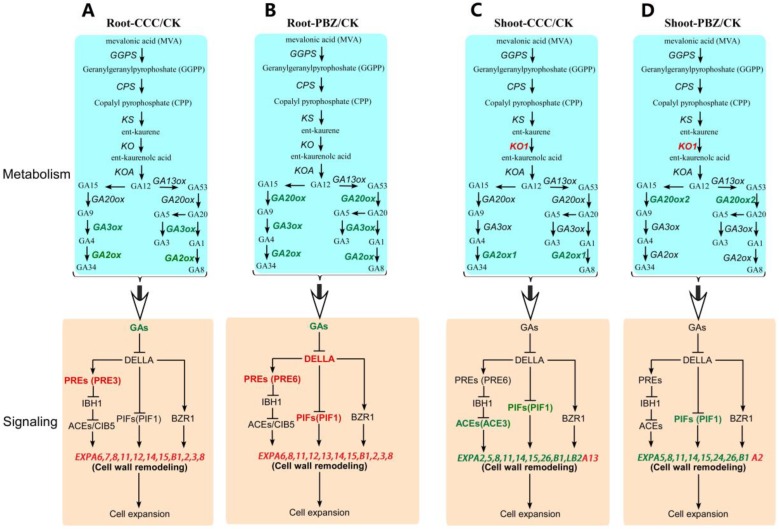
Schematic plot of significantly differentially expressed genes involved in GA metabolism and signaling under CK, CCC, and PBZ treatments in roots and shoots. The plant growth retardants CCC and PBZ restricted the gene expression in the GA metabolism pathway both in the root and shoot in (**A**)–(**D**), and PBZ/CCC treatment raised the gene expression in the GA signaling pathway in roots in A and B, but decreased them in shoots in (**C**,**D**). The genes with red color indicated upregulation, and the green color genes were down regulated.

**Figure 5 ijms-20-03307-f005:**
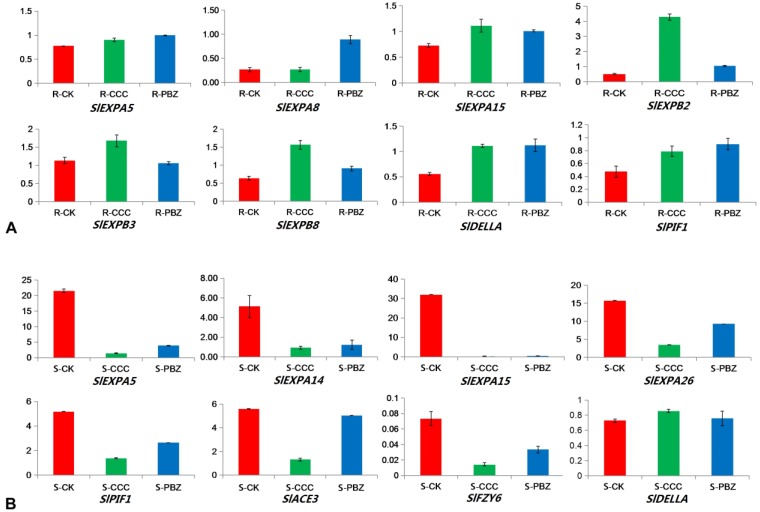
Validation of differentially expressed genes caused by CCC and PBZ. **A**. Eight genes of *SlExpA5*, *A8*, *A15*, *B2, B3*, *B8*, *SlDELLA,* and *SlPIF1* were selected to validate the transcriptome data by qRT-PCR in the root. **B**. In the meantime, in the shoot, eight differentially expressed genes were selected to confirm the RNA-seq dataset: *SlExpA5*, *14*, *A15*, *A26*, *SlPIF1*, *SlACE3*, *SlFZY6,* and *SlDELLA*.

**Figure 6 ijms-20-03307-f006:**
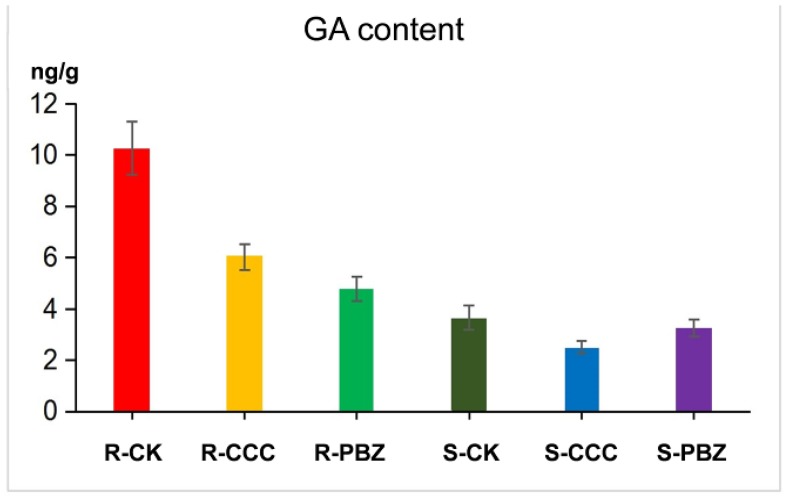
Comparison of gibberellin (GA) content in roots (R) and shoots (S) under CK, CCC, and PBZ treatments. The plant growth retardants CCC and PBZ reduced the endogenous GA content of tomato seedlings both in the roots and shoots.

**Figure 7 ijms-20-03307-f007:**
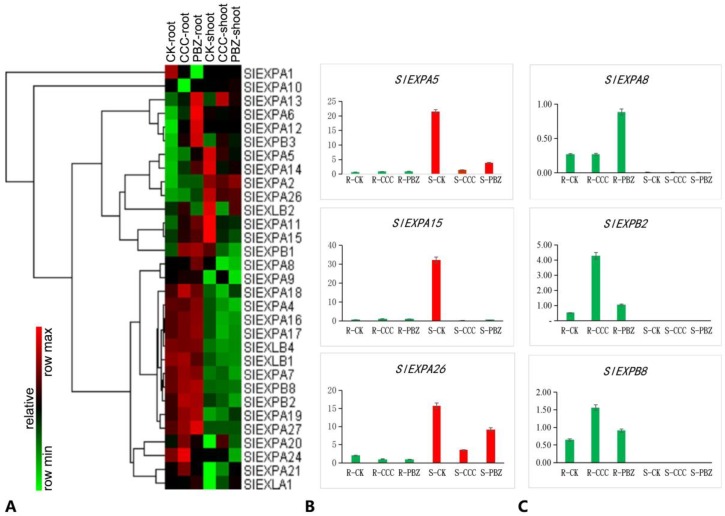
Tissue-specific expression of expansin genes under CK, CCC, and PBZ treatments. **A**. The transcriptome data indicated that plant growth retardants induced an upregulation of root-specific expansin genes like from *SlEXPA8* to *SlEXLA1*, while showing a downregulation of shoot-specific expansin genes such as from *SlEXPA1* to *SlEXPB1*. The color scale on the left shows an increasing expression level from green to red. **B**. Three shoot-specific expansin genes *SlExpA5*, *A15*, and *A15* were confirmed by qRT-PCR. **C**. Three root-specific expansin genes *ExpA8*, *B2*, and *B8* were validated by qRT-PCR technology.

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
