# Peer review of "Comprehensive Transcriptome Reveals an Opposite Regulatory Effect of Plant Growth Retardants in Controlling Seedling Overgrowth between Roots and Shoots"

_ijms, 2019, doi:10.3390/ijms20133307_

Reviewer 1 Report

Dear Editor/Authors,

In the presented article, the Authors present research aimed at demonstrating opposite mechanisms in the roots and shoots during plant growth under the retardants treatments, basing on the transcriptome profiles. In particular, they focus on the role of gibberellin in this process. The information that GA biosynthesis genes are differentially expressed among different tissues (roots, shoots), cell types, and developmental stages is known. In the last few years, many articles have appeared on this subject, also review. Authors should introduce this newest literature in yours work. This will allow them to better prepare a Discussion that is not really a discussion, but mainly a repetition of the results. And the Conclusion is not a conclusion but a short summary.

The applied analytical technique does not raise any objections, however, the preparation of biological material (no data regarding the temperature, light, photoperiod, only 5 replications for each system and the study in only one genotype) suggest that the obtained data cannot be considered as a universal relationship.

Moreover, the picture of plants and the growth parameters calculated presented in Table 1 are the presentation of the same data - the Authors should decide either Fot. or Tab

I suggest shortening the article and presenting the data as a “short article”. Improve the discussion and present it with the results. Add the necessary data regarding plant breeding conditions.

.

Author Response

Dear reviewer, thank you for your professional comments, and we made a point-by-point revision here.

We made an extensive editing of language according the recommendation of MDPI, please see the attachment evidence of editing (English editing ID: English-10490).

Please check the point-by-point revision as follow:

1. Authors should introduce this newest literature in yours work.

Response: Thanks, we added several references in this new edition of manuscript. In addition, we revised the first and second paragraph of the introduction part, which emphasized the importance of plant growth retardants in controlling seedling overgrowth, especially in the water cultured system, to meet the rapid development of seedling raising in recent years.

2. Discussion revision suggestion.

Response: We restricted and revised the discussion part in the new manuscript. It was composed of three key topics that the plant growth retardants restrained GA biosynthesis in controlling overgrowth, regulated GA signaling in an opposite way between roots and shoots, and controlled overgrowth through the tissue-specific SlEXPs genes.

3. Conclusion revision suggestion.

Response: We've modified the conclusion as follow:

The plant growth retardants PBZ and CCC were used to control overgrowth in water-cultured tomato seedlings, with concentrations of 1 ppm and 20 ppm, respectively. The plant growth retardants PBZ and CCC could restrain the expression of GA synthesis and signaling gene expression, but with an opposite regulation between the shoots and roots. Among the DEGs related to the GA pathway, the SlPIF1 and the downstream genes SlEXPs were revealed to be key regulators in controlling overgrowth in tomato seedlings.

4. No data regarding the temperature, light, photoperiod.

Response: Thank you very much, we added the condition of seedling cultivation in the new manuscript of line 321-322. Please check it.

5. Table 1 are the presentation of the same data with Figure 1.

Response: Modification as suggested, we move the table 1 to supplementary table 1, Thanks a lot.

In addition, we simplified the title in the new manuscript as ‘Comprehensive transcriptome reveals an opposite regulatory effect of plant growth retardants in controlling seedling overgrowth between roots and shoots’, it was easier to understand for global readers of IJMS.

And we revised the methods information in line 315-316 and line 322-324, including information of variety test and concentration of PBZ/CCC derived from the pre-experiment. In the pre-experiment study, we test several pink tomato varieties in China, the phenotypic effect of PBZ/CCC was similar each other. Yingfen No. 8 was chosen to be analyzed in transcriptome sequencing in this study.

Dear reviewer, we really appreciate for your professional review.

Have a nice day.

Best regards

Reviewer 2 Report

The manuscript ID # ijms-522547 titled "Comprehensive transcriptome profile reveals opposite regulatory mechanism between in root and shoot under plant growth retardants treatment in controlling seedling overgrowth" by Li et al. described by phenotypic analysis and mainly by transcriptomics comparison how the main plant growth retardants control the seedling overgrowth in tomato. I found the manuscript rather well written (some minor english style check should be carried out) and the experimental design appropriated to reach the goal of the Authors. 

Therefore, the overall scientific message that the paper brings home appeared fairly obvious to me. The GA pathway has been studied deeply in the last century and the role of related pathway genes in the elongation of shoot tissues is not really a novelty. Also the GA-related genes different role between root and shoot has been previously highlighted in many papers.

Finally, I did not understand the meaning of the conclusion, what do these results suggest to the Authors and to the readers? I would have considered most interesting a comparison between two (or more) tomato genotypes with different behavior about seedling overgrowth to understand how the retardants application could be reduced. As alternative, on the same genotype the Authors should have apply different growth condition (light, nutrient and others) in order to understand the modification on the plant growth retardants application. 

Thus, I suggest that the Authors will reconsider their results that appeared to me scholastic and obvious in light of a large revision of their discussion and mainly the conclusion and future perspective for tomato growers.           

Author Response

Dear reviewer, thanks a lot for your professional review, and we made a point-by-point revision based on your comments.

Dear reviewer, Firstly, we made an extensive editing of language according the recommendation in the MDPI system, please see the attachment evidence of editing (English editing ID: English-10490).

Secondly, the point-by-point revision was as follow:

1.Conclusion revision suggestion.

Response: Thanks, we revised the conclusion as follow:

The plant growth retardants PBZ and CCC were used to control overgrowth in water-cultured tomato seedlings, with concentrations of 1 ppm and 20 ppm, respectively. The plant growth retardants PBZ and CCC could restrain the expression of GA synthesis and signaling gene expression, but with an opposite regulation between the shoots and roots. Among the DEGs related to the GA pathway, the SlPIF1 and the downstream genes SlEXPs were revealed to be key regulators in controlling overgrowth in tomato seedlings.

2.Comparison between two (or more) tomato genotypes.

Response: Thanks. In the pre-experiment study, we test several pink tomato varieties in China including the Yingfen No.8, the phenotypic effect of PBZ/CCC was similar each other (unpublished data). The shoot and root were restricted by treatment of PBZ/CCC. Because of the Yingfen No. 8 is one of the most typical pink tomato varieties, of which is the majorly cultivated in China, it was chosen to be analyzed in transcriptome sequencing in this study.

3. Apply different growth condition (light, nutrient and others)

Response: Revision made as suggested; we added the condition of seedling cultivation in the new manuscript of line 321-322. The tomato seedlings were cultivated in the greenhouse at a photoperiod light/dark of 16/8 h, with temperatures during the day/night of 28–32/18–22℃.

4. Discussion revision requirement.

Response: We restricted and revised the discussion part in the new manuscript. It was composed of three key topics that the plant growth retardants restrained GA biosynthesis in controlling overgrowth, regulated GA signaling in an opposite way between roots and shoots, and controlled overgrowth through the tissue-specific SlEXPs genes.

In addition, we simplified the title in the new manuscript as ‘Comprehensive transcriptome reveals an opposite regulatory effect of plant growth retardants in controlling seedling overgrowth between roots and shoots’. And we revised the first and second paragraph of the introduction part, These revisions made the current manuscript easier to understand for global readers of IJMS.

Dear reviewer, we really appreciate for your professional review.

Have a nice day.

Best regards

Round  2

Reviewer 1 Report

Dear Editor / Authors,
all comments have been included in the current version.
The article has been corrected and I am recommending the version presented for publication
I wish you further scientific successes

Author Response

Thank you for your comments.

Reviewer 2 Report

I found the new manuscript version with the same trivial message to the readers although the Authors produced many efforts in the work revision. I found the answer point by point letter not completely exhaustive of the reviewer suggestions and request of modification. The authors answer about a pre-experiment study in which they tested several pink tomato varieties in China observing similar effects of PBZ/CCC treatments among genotypes confirmed my idea that the results are not of particular interest. It would be more interesting if a genotype showed a different natural behavior for overgrowing regardless the treatments. 

Most importantly I wrote "As alternative, on the same genotype the Authors should have apply different growth condition (light, nutrient and others) in order to understand the modification on the plant growth retardants application". 

The answer "we added the condition of seedling cultivation in the new manuscript of line 321-322. The tomato seedlings were cultivated in the greenhouse at a photoperiod light/dark of 16/8 h, with temperatures during the day/night of 28–32/18–22℃" is not exactly what I suggest to discuss by the Authors.

Finally, in the new version of the conclusion I did not find any interesting future perspective for tomato genetics and for growers.

Author Response

Dear reviewer, Thank you very much for your comments about our experiments and the conclusion. We’d like to explain more about these two parts, which was poorly responded in the last time. Please take your valuable time to check the new response, and considerate the utilization value of our experiments in practical production. Our group really appreciates for your professional review.

Dear reviewer, please check the two point-by-point responses as follow:

  1. About the experiment in method part (Apply different growth condition (light, nutrient and others).

Response: Thank you very much. Dear reviewer, we reported the utilization of plant growth retardants PBZ/CCC in controlling the overgrowth of tomato seedlings, in the water culture system in this study. Compared with other cultivation system, it had many pros of saving resources in the water culture system, thus it was in raising used fast in seedling industries of China recently. There were more than 3 billion tomato seedlings traded in the Chinese market every year. However, the overgrowth of tomato seedlings was one of the most important limitations for seedlings nursing in industries, and the industries had the regular cultivation condition of tomato seedling, of which was cultivated in greenhouses with the photoperiod light/dark of 16/8 h, and temperature of day/night of 28–32℃/18–22℃, and the same nutrient solution of 5 mM KNO3, 5 mM Ca(NO3)2, 2 mM MgSO4, 1 mM KH2PO4, 50 M FeNa2(EDTA)2, 50 M H3BO3, 10 M MnCL2, 0.8 M ZnSO4, 0.4 M CuSO4, and the 0.02 M (NH4)6MoO24. This cultivation condition was proved as be the most effective way in tomato seedling nursing, and was supported by Beijing Vegetable Research Center, Beijing, China. About the tested varieties in this study, we tested several pink tomato varieties to control the seedlings overgrowth by treating PBZ/CCC in the pre-experiment study, including the Xianke 5, Zhefen 702, Shenfen 998, and the Yingfen No.8. These pink tomato varieties were uniformly restricted in shoots and roots with slight differences because of close genetic background probably. The pink tomato varieties were extensively planted in China, more than 50% of cultivations, which were attracted great attentions of seedling industries. For the transcriptome experiment, we selected the variety of Yingfen No.8 randomly, of which was set up 12 treatments of PBZ/CCC and control, with excellent sequencing quality in this study. It was not easy to sequence more varieties in the same treatment of PBZ/CCC, as well as the same varieties with different cultivation conditions. Because of that the variety sequenced was well responsive in Chinese market and the cultivation condition was regular and effective in the seedlings nursing industries.

2. About the conclusion.

Response: Thanks a lot. In this study, we confirmed that the plant growth retardants PBZ/CCC restricted the GA biosynthesis, which was well studied in previous reports, in the water culture system using the effective concentration of 1 and 20 ppm, respectively. Moreover, we found that the signaling pathway genes of GA were significantly affected, and behaving in an oppositely regulation pattern between roots and shoots. Additionally, the SlPIF1 and several target candidates of tissue specifically expressed SlEXPs were observed in responsive to plant growth retardants PBZ/CCC in tomato seedlings. And they could be acting as marker genes in response to overgrowth in tomato seedlings, as well as the treatment of plant growth retardants PBZ/CCC, of which was important to be used to select key varieties in the tomato breeding system. In recent months, we are working on the transgenic validation of these marker genes obtained in this study. Dear reviewer, we hope that our responses would meet your questions, please give your extensive comments about our manuscript, and we’d like explain more about it based on your professional judgments. At last, please considerate the utilization value of our experiments in practical production. Have a nice day, Best regards, Yours Sincerely, Prof. Mingchi Liu Prof. Changlong Wen